# In Situ Block Size Distribution Aimed at the Choice of the Design Block for Rockfall Barriers Design: A Case Study along Gardesana Road

**Gessica Umili [1],\*** , **Sabrina Maria Rita Bonetto [1]** , **Pietro Mosca [2]**, **Federico Vagnon [1]** and **Anna Maria Ferrero [1]**

1   Department of Earth Sciences, Università degli Studi di Torino, via Valperga Caluso 35, 10125 Torino, Italy; sabrina.bonetto@unito.it (S.M.R.B.); federico.vagnon@unito.it (F.V.); anna.ferrero@unito.it (A.M.F.)
2   Institute of Geosciences and Earth Resources—CNR, via Valperga Caluso 35, 10125 Torino, Italy; pietro.mosca@cnr.it
\*   Correspondence: gessica.umili@unito.it; Tel.: +39-011-0918364

**Abstract:** When studying rockfall phenomena, a single value of the block volume is not sufficient to take into account the natural variability of the geometrical features (orientation, spacing, persistence) of the discontinuity sets. Different approaches for obtaining cumulative distributions of potentially detachable block volumes are compared. A highly fractured rock mass outcropping along the western Lake Garda (Italy), consisting of prevailing limestone and interbedded marls, is studied in detail from geological and geostructural points of view. Then, a representative rock face has been selected and analyzed with traditional and non-contact survey methods to identify the main discontinuity sets and to collect spacing samples. Based on these data, in situ block size distributions for different combinations of sets are built following statistically-based approaches, without the use of a Discrete Fracture Network (DFN) generator. The validation of the obtained distributions is attempted based on the detached block surveyed at the foot of the slope. However, in this particular case study, the detached blocks cover only a minimal volume range compared to both theoretical values and visible rockfall scars. The fallen rock blocks have a marginal role in design block determination, since their volume depends on geological discontinuities (bedding and fractures) and could be affected by other processes after the detachment (e.g., fragmentation). The procedure here described should be standard practice in the study of rockfall events, and it should be uniform in European standards such as Eurocodes. Future developments should involve the scientific community for setting the percentiles of the probability distribution to be considered for block design definition.

**Keywords:** rockfall; spacing; block volume; IBSD; Lake Garda

## 1. Introduction

The rockfall phenomenon is defined as the detachment, fall, rolling, and bouncing of rock fragments, which can break during impacts [1]. Events range from small pebbles falling down a cliff to massive boulders of several tonnes rolling down a slope [2]. Rockfalls represent one of the major causes of fatalities associated with landslides due to the high energy and mobility of the involved rock blocks [3]. Rockfall modeling aims to assess the envelope of trajectories, the maximum runout distance, the distribution of kinematic parameters along a fall path, and the probability for a specified "design block" to stop at specific distances from the starting point [4].

However, a single value for the design block is not sufficient to take into account the natural variability of the geometric features (orientation, spacing, persistence) of the discontinuity sets.

A cumulative distribution of block volumes that could be potentially generated by discontinuity sets in a rock mass is the correct approach.

The prediction of in situ block size distribution (IBSD), or better, its assessment, since the true IBSD can rarely be evaluated, has been one of the primary pursuits of mining and quarrying operations [5]. Size and shape determinations of in situ rock fragments are becoming increasingly important issues in materials mining, comminution and handling industries, and in the construction industry using stone products [6].

Many methodologies were developed to forecast IBSD. For instance, Wang proposed "the Equation Method", which is based on a formula and a series of look-up tables [7], to define the cumulative curve for the in situ block sizes, considering negative exponential, log-normal, and uniform distributions of spacing values. The equation method uses a set of empirical equations to estimate the IBSD, relating it to the mean spacings and the mean orientations of the three primary sets of discontinuities. The equations are obtained using the exact dissection method solutions described in [8], based on the block theory by Goodman and Shi [9].

Elmouttie and Poropat [10] proposed a new method to predict IBSD in fractured rock masses using realistic DFN, robust polyhedral modeling, and a Monte Carlo sampling approach, to account for the stochastic variability in the fracture geometry. This approach is promising but not directly applicable to survey data without the use of a DFN generator.

Stavropoulou [11], following the methodology proposed originally by Priest and Hudson [12], introduced the closed-form expression of block volume cumulative distribution based on joint frequencies measured along drilled cores. This method initially developed for the quarry environment can hardly be transposed to natural slopes.

In 2006, Latham and coauthors [13] stated that, due to the rapid advances in digital imaging and semi-automated geological face mapping, software packages devoted to rock mechanics applications might soon enable IBSD to be rapidly determined from software, with a limited manual mapping of joints. Nowadays, non-contact survey methods have indeed become more robust, producing reliable joint orientation [14–26], trace length [27–29], and spacing data [30,31]. However, IBSD from survey data is not yet an option in rock mechanics codes for the study of rockfall phenomena, even though IBSD could provide useful tools for the design of flexible barriers.

Buyer and coauthors [32] presented a consistent and semi-automatic process for rock mass characterization, starting from photogrammetry and ending in numerical modeling. The Discontinuity Set Extractor [23], a semi-automatic method for identifying and analyzing the surfaces of rock outcrops, is used in combination with the software SMX Analyst (3GSM GmbH). Then, the joint network characteristics are used in 3DEC (The Itasca Consulting Group) to determine the IBSD and shape distribution at a very high level of detail and objectiveness. However, this level of automation is far from available to engineers who deal with the design of rockfall protection barriers.

In order to follow the requirements of the Limit State Design (LSD), as defined by Eurocode 7 (EC7) [33], survey methods should guarantee a high level of accuracy and precision of the results to reduce the uncertainties of the estimate of the design parameters. In essence, this is particularly relevant for the geometric characteristics of discontinuities, since the dimension and shape of potentially detachable blocks depend on their values [34].

Ferrero and coauthors [35] proposed a simple application of interval analysis to spacing data obtained from non-contact surveys for block volume estimation to quantify the influence of uncertainty on spacing and relative orientation. Since in rockfall phenomena the involved kinetic energy is strongly related to the block mass, the structural capacity of flexible barriers is defined based on the characteristic block volume, which at present is often subjectively decided by the designer. For example, EC7 [33] gives no thorough information concerning the choice of the design block; only national standards, such as UNI 11211-4 in Italy [36] and ONR 24810 in Austria [37], provide suggestions on its definition. In Italy, the characteristic block volume is defined based on the experience and knowledge acquired after an onsite survey. Conversely, ONR standards propose a percentile of the IBSD in relation to three



consequence classes defined in EN 1990: 2002 [38]. However, the commonly assumed 95% fractile (or higher) value for IBSD would be very restrictive and improbable for many rockfall events [39].

A statistically robust IBSD represents a promising tool for a more practical choice of the design block, mainly because it covers the entire volume range. Therefore, the designer could take advantage of IBSD for comparing different scenarios concerning the technical features of the rockfall barrier, locations, and costs.

This work aims to compare different methods to infer IBSD, considering a rock mass outcrop in the Gargnano-Muslone area along the western Lake Garda (Brescia province, Italy; Figure 1). The investigated outcrops, consisting of prevailing limestone and interbedded marls, form steep rock faces along the main (Gardesana Road) and secondary roads and record continuous rockfall phenomena due to their high degree of fracturing.

This work also aims to bring attention to the possible limitations of the application of methodologies for estimating the return period of rockfall blocks according to their size (e.g., [40]). The survey of blocks accumulated at the slope foot and the available historical data could be insufficient, particularly in rock masses characterized by a complex geostructure as the one described here. Moreover, during impact with the ground, the detached rockfall mass can undergo fragmentation along pre-existing discontinuity planes, by the propagation and coalescence of discontinuities into rock bridges, or by the development of new cracks in intact rock [41]. The validation of the obtained curves is attempted, showing the limited contribution of data acquired at the foot of the slope in this particular case.

## 2. Geological Setting

The Lake Garda region belongs to the Southern Alps (Figure 1A), a structural domain of the Alps, and the western side of the lake consists of Mesozoic lithologies of the Lombardian Basin.

The Southern Alps identify a south-vergent fold-and-thrust belt separated from the north-vergent collisional belt by the Periadriatic Lineament and preserve part of the Mesozoic continental margin of the Adriatic plate (e.g., [42,43]). The Lombardian Basin developed as a complex subsiding area during Early Jurassic on the previously weakened crust and was separated to the east from the Venetian Platform by the extensional Ballino-Garda fault [44]. During the Alpine orogenesis, the Lombardian Basin succession was deformed by the southward propagation of the Southern Alps belt [45].

In the present structural framework of the Southern Alps, the western Lake Garda lies between the Tremosine–Tignale thrust system and the inverted Ballino–Garda fault (Figure 1B). The exposed sedimentary succession is part of the Lombardian Basin, which is tectonically overlain by Norian platform carbonates to the west [46–48]. Massive platform limestones of the Corna Formation (Rhaetian–Early Jurassic p.p.) are upsection followed by Lower Jurassic pelagic carbonates (mainly cherty limestone and marly limestones) of the Medolo Formation. The Middle Jurassic Concesio Formation (marlstone and marly limestone) is overlain by siliceous sediments of the Selcifero Lombardo Formation of Middle-Late Jurassic age, which is in turn covered by thinly bedded limestones of the Maiolica Formation (Early Cretaceous in age). The upper stratigraphic levels consist of very thinly stratified calcareous and argillaceous marls of the Scaglia Lombarda Formation (Late Cretaceous—Early Eocene).

Along the western Lake Garda, the whole Lombardian Basin succession is folded by km-scale anticlines and synclines (Figure 1B), and in the investigated Gargnano-Muslone area, the core of the main syncline consists of the Scaglia Lombarda Formation [48]. In this area, open to tight asymmetrical folds (Figure 2A) have mainly NE–SW trending axes and axial planes variably dipping toward NW.

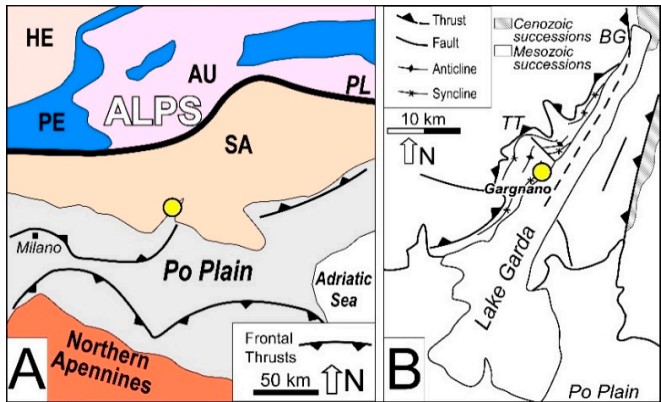

**Figure 1.** (**A**) Tectonic map of the central part of the Alps (modified after [49]). North-vergent collisional belt (North of Periadriatic Lineament): Austroalpine (AU), Helvetic–Dauphinois (HE), Penninic (PE). Southern Alps (SA) rest South of Periadriatic Lineament. Periadriatic lineament (PL) and fronts of buried thrusts in the Po Plain are shown. (**B**) Structural setting of the Lake Garda region: (TT) Tremosine–Tignale and (BG) Ballino–Garda fault systems (modified after [49]). The yellow dots in (**A**,**B**) indicate the study area.

In this geological setting, the analyzed rock mass is part of the Medolo Formation (Figure 2B,C), defining the southern limb of the Gargnano-Muslone syncline. This formation is mainly made of marly limestone and cherty limestone forming 10–40 cm thick beds and thin-bedded gray marls characterized by a thickness usually ranging from a few centimeters to 30–40 cm but locally significantly thicker. The bedding of the investigated outcrops dips toward NW and W at a moderate angle. In the area, mesoscopic compressional structures (top-to-SE) subparallel to the bedding have been observed.

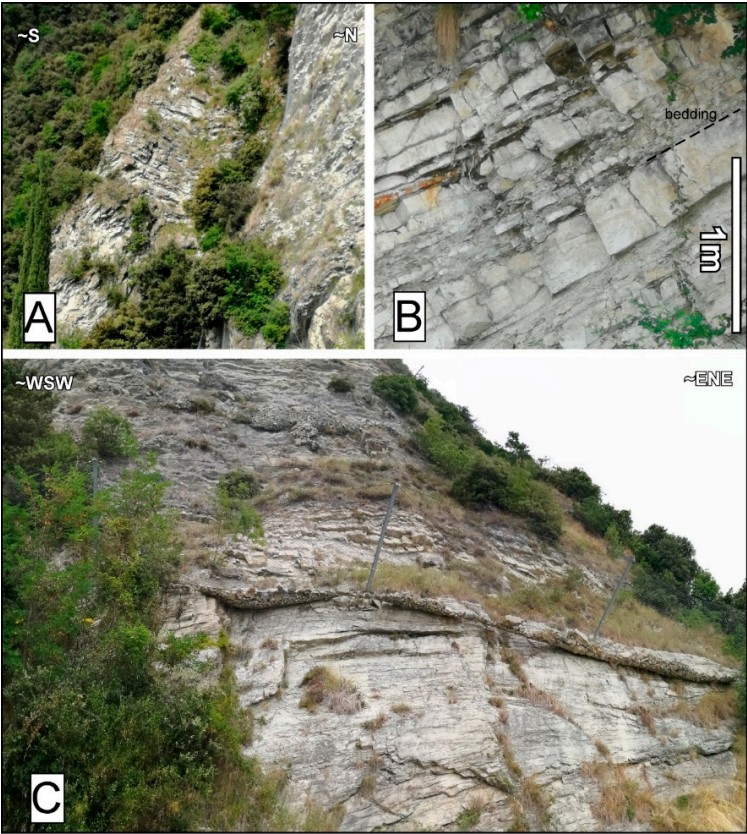

**Figure 2.** (**A**) Folds deforming the Jurassic succession. (**B**,**C**) Details of the marly-limestone multilayer in the Medolo Formation.

### 3. Methods

This section focuses on the block volume assessment. It aims to highlight the transition from the calculation of a deterministic value to the construction of different IBSDs based on the statistical processing of spacing samples. Survey methods are also included to complete the description.

*3.1. Block Volume Assessment*

Miles [50] proposed analytical equations to calculate the mean and variance of volume V of the blocks, which are created either by three discontinuity sets with a negative exponential spacing probability distribution (regular subdivision) or by a random space partition with planes placed by a Poisson process with uniform density (random subdivision). The comparison between the two subdivisions shows that the random one generates volume that is on average bigger than those created by the regular subdivision (ratio of about 2).

Palmstrøm [51] provided guidelines for estimating the block volume from various types of joint density measurements. His well-known equation in the case of 3D measurements of three discontinuity sets generating a block is:

$$V_B = \frac{S_1 S_2 S_3}{\sin \gamma_{12} \sin \gamma_{23} \sin \gamma_{13}}, \tag{1}$$

where $S_1$, $S_2$, and $S_3$ are the spacing of the three sets of discontinuities; $\gamma_{12}$ is the angle between set 1 and set 2, and similarly for $\gamma_{23}$ and $\gamma_{13}$.

These methods have some limitations: first of all, input values (frequency or spacing of discontinuity sets) are intended as representative values, and this often results in a rough calculation of the average, without considering the actual statistical distribution of the sampled data; besides, the output is a deterministic value, and this is far from representative of a complex medium as a rock mass.

*3.2. IBSD Assessment*

3.2.1. Method Based on Empirical Coefficients

IBSD constitutes an evolution of block volume calculation: it represents the cumulative curve of the potentially detachable in situ blocks, and its construction considers the frequency distributions of spacing values. A particularly simple method for IBSD assessment was proposed by Lu and Latham [5]. They revised the equation method [7] and added some refinements, including the fractal spacing distribution and the influence of persistence. They obtained the following equation, describing the volume of blocks generated by three sets of discontinuities:

$$V_{i,p} = \frac{C_{i,p}}{\left(F_{imp}\right)^q} \frac{S_{pm1} S_{pm2} S_{pm3}}{cos\theta cos\phi cos\alpha} \quad i = 10, \ 20, \ \ldots, 100, \tag{2}$$

where $S_{pm1}$, $S_{pm2}$, and $S_{pm3}$ are the principal mean spacing values, namely the average true spacing values of the three sets; $\theta$, $\phi$ and $\alpha$ are the angles between the mean orientations of the three sets; and $q$ is a constant less than 1, whose value can be assumed between 1/5 and 1/2 [52]. One should note that since the angle between a pair of sets can theoretically vary from 0° to 180°, its cosine assumes a negative value for an angle >90°: however, since $cos(\pi - x) = -cos(x)$, it is sufficient to consider always the absolute value of the cosine.

The relative "impersistence factor" $F_{imp}$ [52] is defined as follows:

$$F_{imp} = \begin{cases} \frac{S_D}{S_r} & S_D < S_r \\ 1 & S_D \geq S_r \end{cases}, \tag{3}$$

where $S_D$ is the mean discontinuity size. It can be estimated under different assumptions about the discontinuity shape (circular, elliptical) and using different methods. In particular, methods for estimating the mean discontinuity diameter E(D) based on the mean trace length E(l) provide formulations that can be applied to trace lengths sampled in circular windows (a review of these methods is in [53]). *Sr* represents the characteristic size of the rock mass under consideration, i.e., the cube root of the volume of the rock mass of interest.

$C_{i,p}$ is an empirical coefficient: values for discontinuity sets with negative exponential and uniform distributions, as well as with a log-normal distribution spacing law [54], are summarized in Table 1. Lu and Latham [5] added the values for a rock mass for which the three sets of discontinuities have fractal spacing distributions (Table 1). The range is the standard deviation multiplied by a constant of 1.64. The 90% confidence interval for $C_{i,p}$ can be derived from $C_{i,p} \pm$ Range.

**Table 1.** Coefficients $C_{i,p}$ with 90% confidence intervals for the relationship in Equation (2), considering different distributions of the product of principal mean spacing values of discontinuities (modified from [5]).

| Passing | Uniform | | Negative Exp. | | Log-Normal | | Fractal | |
|---|---|---|---|---|---|---|---|---|
| % | $C_{i,p}$ | Range | $C_{i,p}$ | Range | $C_{i,p}$ | Range | $C_{i,p}$ | Range |
| 10 | 0.375 | 0.157 | 0.332 | 0.131 | 0.469 | 0.099 | 0.4649 | 0.0128 |
| 20 | 0.700 | 0.292 | 0.710 | 0.249 | 0.965 | 0.207 | 1.1685 | 0.0409 |
| 30 | 1.052 | 0.435 | 1.207 | 0.423 | 1.513 | 0.334 | 2.1606 | 0.0748 |
| 40 | 1.460 | 0.607 | 1.852 | 0.645 | 2.220 | 0.542 | 3.5458 | 0.1612 |
| 50 | 1.939 | 0.787 | 2.708 | 0.984 | 3.099 | 0.731 | 5.3165 | 0.192 |
| 60 | 2.548 | 1.036 | 3.980 | 1.550 | 4.287 | 1.029 | 8.0903 | 0.3098 |
| 70 | 3.343 | 1.384 | 5.867 | 2.596 | 5.956 | 1.501 | 13.3920 | 0.8398 |
| 80 | 4.495 | 1.802 | 8.948 | 4.581 | 8.497 | 2.243 | 22.6070 | 2.2649 |
| 90 | 6.623 | 2.691 | 15.332 | 9.532 | 13.377 | 4.227 | 39.6660 | 5.0295 |
| 100 | 17.772 | 9.348 | 38.992 | 23.734 | 38.277 | 17.569 | 108.9700 | 9.6708 |

In [13], a practical step-by-step methodology for IBSD assessment is reported: it includes approaches that do not rely on specialized computer software. It suggests using actual discontinuity data from traditional and non-contact surveys to apply Wang's equation method.

1. Determine the principal spacing for the three different sets.
2. Determine the three characteristic angles $\alpha$, $\beta$, and $\phi$.
3. Plot the principle spacing data as histograms and assign the appropriate principal spacing distribution. Hence, determine the principal mean spacing.

   Apply Equation (2) using values from Table 1 for assessing the IBSD.

### 3.2.2. Method Based on the Cumulative Distribution Function

As discussed in [11], the spacing of a discontinuity set can be considered a continuous random variable; therefore, its cumulative distribution function (CDF), denoted as $F(x) = P\{X \le x\}$, defines the probability that a given spacing value X is less than x.

The spacing data of each discontinuity set were processed using a Matlab Statistics Toolbox: gamma, negative exponential, log-normal, and Weibull distribution functions were fitted to the data. Their goodness of fit was evaluated through the one-sample Kolmogorov–Smirnov (K–S) test.

The K–S test is a nonparametric test comparing the population CDF of the data and the hypothesized CDF. The null hypothesis is that the data in vector x come from a specific distribution. The test accepts the null hypothesis if the *p*-value is higher than the significance level, here, it is assumed to be equal to 1%.

Once the best fitting distributions were identified for each discontinuity set, the obtained CDFs were input in Equation (1) instead of single spacing values. Regarding the angles between the

considered discontinuity sets, their average values were assumed as the characteristic values, since their influence on volume variability was proven to be limited, compared to that of spacing [35]. The result is the block volume CDF, which in the following will be called Sample CDF (SCDF). Since the equation considers a group of three sets generating a block, different terms were analyzed.

### 3.2.3. Method Based on Monte Carlo Simulation

Based on the best fitting distribution previously obtained for a set, 100,000 input spacing values were randomly generated from that distribution, and the corresponding CDF was obtained. The same was done for all the discontinuity sets. As for the measured spacing samples, CDFs were input into Equation (1) for calculating the block volume CDF. In the following, this will be called Monte Carlo CDF (MCDF).

### 3.3. Surveys for Block Volume Parameters Definition

Due to the morphology of the slope, most of the outcrops were not directly accessible. Therefore, non-contact techniques and traditional geomechanical surveys were applied to define the main parameters for block volume assessment, as described in the previous sections.

### 3.3.1. Non-Contact Survey

Non-contact survey techniques operate on the representation of the area of study through a 3D model made of one or more dense point clouds. In this study, point clouds were obtained by both laser scanner and photogrammetric surveys, which were performed by IMAGEO S.r.l.

The laser scanner Teledyne Optech Polaris LR was used first, setting an average point density of 1 point/cm. Due to the morphology of the rock slope, the laser scanner acquisition suffered from the physical restrictions in the instrument positioning. Therefore, a photogrammetric survey from a drone was also performed. The digital images were acquired with the integrated camera of the system DJI Phantom 4 (focal length 3.61 mm, resolution 4000 × 3000 pixels).

A total of 545 images were shot, with a ground resolution of 6.6 cm/pix, covering an area of about 0.6 km$^2$. The shooting points were aligned horizontally and vertically in routes parallel to the slope; the camera was shot with nadiral and oblique inclination. The minimal overlapping was about 80% among images of the same longitudinal route and about 60% transversally among the routes.

Laser scans and digital images were georeferenced based on Ground Control Points (GCP). GCPs were acquired by GPS differential technique using GNSS Receivers Topcon Hiper V, referring to the GNSS permanent station of Rovereto (TN). Software Agisoft Metashape was used for performing point cloud reconstruction based on digital images. Then, point clouds were merged to create a Digital Surface Model (DSM) of the entire slope.

Software Rockscan [18] was used to manually delimit discontinuity planes on the images and estimate their dip and dip direction. This method allowed surveying 1580 discontinuity planes from which the main sets were identified.

Software CloudCompare was used to measure spacing among planes belonging to the main sets. Tools included in Compass plugin were used; the user manually drew a line perpendicular to the discontinuity traces belonging to the same set and measured their relative distances, avoiding the need for corrections of the Terzaghi's bias.

### 3.3.2. In Situ Traditional Survey

In the accessible area at the base of the slope, a geomechanical survey campaign was carried out according to the International Society for Rock Mechanics (ISRM) suggested methods [55]. Traditional surveys along scanlines were performed to cense all the discontinuities intersecting the scanlines and describe geometric and kinematic features, such as orientation, spacing, persistence, roughness, wall strength, aperture, degree of weathering, seepage, presence, and nature of filling.

This method made it possible to conduct the survey of 138 planes. The information on the type of discontinuity, number, and orientation of the main sets helped in the validation of the non-contact survey data.

### 3.4. Blocks Volume Survey at the Foot of the Slope

The Rockfall Block Size Distribution (RBSD) introduced by Ruiz-Carulla and coauthors [56] refers to the rockfall deposit. Obtaining the RBSD of a large rockfall deposit may become a challenge due to the high number of blocks to be measured. The methodology was to lay a tape measure along the blocks deposit at the foot of the slope. By walking along the tape measure, a sequence of images of the deposit was shot: in each image, a portion of the tape measure was visible. It was used to set the scale of the image content utilizing ImageJ, which is an open-source image processing program for multidimensional image data, with a focus on scientific imaging. Then, in each image, we selected blocks with three visible edges and measured them: sizes of 394 blocks were acquired. Due to the heterogeneity of block shapes, two geometrical solids were assumed to calculate block volume: parallelepiped (Equation (4)) and ellipsoid (Equation (5)).

$$V_{par} = L_1 L_2 L_3, \tag{4}$$

$$V_{ell} = \frac{4}{3}\pi \frac{d_{12}}{2}\frac{d_{13}}{2}\frac{d_{23}}{2} = \pi \frac{d_{12}d_{13}d_{23}}{6}, \tag{5}$$

where $L_{1,2,3}$ are the lengths of three different edges of the block, and $d_{1,2,3}$ are the diagonals of three different faces, which are calculated as

$$d_{ij} = \sqrt{L_i^2 + L_j^2}. \tag{6}$$

Surveyed volumes will be referred to as $SV_{par}$ and $SV_{ell}$ to distinguish between the two assumed shapes.

## 4. Application to the Case Study

Traditional and non-contact geomechanical surveys were carried out on a representative slope in the Medolo Formation (see the yellow dot in Figure 1) interested by rockfall events. The results are reported in the following sections.

### 4.1. Traditional Survey

Data collected along three scanlines arranged in correspondence of accessible areas at the foot of the slope indicate the presence of six main orientations for the discontinuities, which are schematically represented in the lower hemisphere plot of Figure 3.

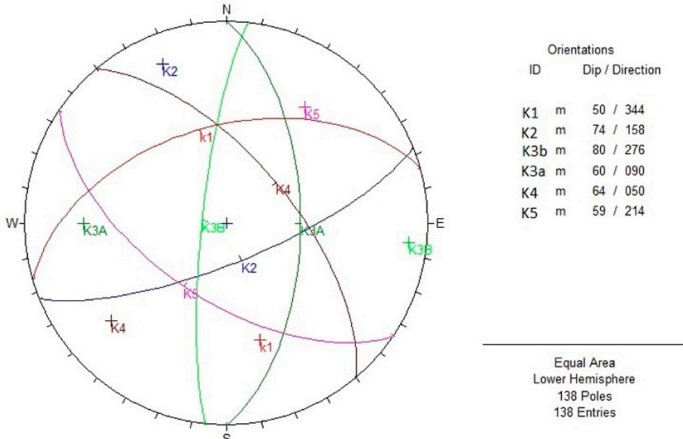

**Figure 3.** Plot reporting the main discontinuity sets obtained by processing scanlines data.

Considering the features observed on the field and the geological knowledge of the area, they were grouped into five main sets (labeled as K) as follows (Figure 4).

K1 is the bedding. It dips mainly toward NW at a medium-high angle and can be southward-dipping on the short limb of asymmetric folds. It is persistent and regularly presents itself along the scanlines with centimetric to decimetric spacing.

Discontinuities dipping SE at high angles consist of fractures indicated as K2. They have been identified in all scanlines and have decimetric spacing.

Fractures E and W-dipping at a high angle were observed and associated with the same set; they are named K3a and K3b, respectively. At the outcrop scale, they are regularly present and show low persistence and centimetric to metric spacing.

Besides, two sets of fractures were occasionally surveyed, dipping toward NE and SW at a medium-high angles, named K4 and K5, respectively. They show high persistence.

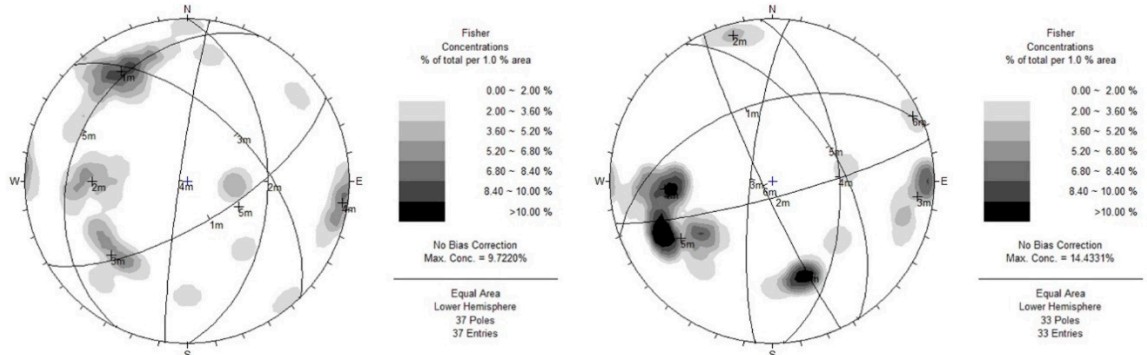

**Figure 4.** Poles plot of two different surveys along scanlines.

Table 2 reports the average orientation and spacing range of the main discontinuities sets obtained from traditional surveys.

**Table 2.** Average orientation and spacing range of the main discontinuities sets obtained from traditional surveys.

| Set | Dip | Dip Direction | Spacing Range (m) |
|-----|-----|---------------|-------------------|
| K1 | 50 | 344 | 0.02–1.55 |
| K2 | 74 | 158 | 0.22–3.38 |
| K3a | 60 | 090 | 0.02–2.88 |
| K3b | 80 | 276 | 0.07–3.44 |
| K4 | 64 | 050 | 0.18–1.86 |
| K5 | 59 | 214 | 0.15–0.96 |

### 4.2. Non-Contact Survey

Due to the large extension of the rock face, the non-contact survey allowed for the acquisition of a large number of data, assuring higher representativeness of the measurements at the slope scale.

The rock face was divided into different windows, respectively named 3, 3A, 3B, 3C, 3D, 3E, 3F, and 3G (Figure 5). The DSM of each of them was input in Rockscan, along with the corresponding images. A high number of discontinuities were recognized and described in terms of orientation (dip/dip direction) and grouped into sets (Figure 6 and Table 3).

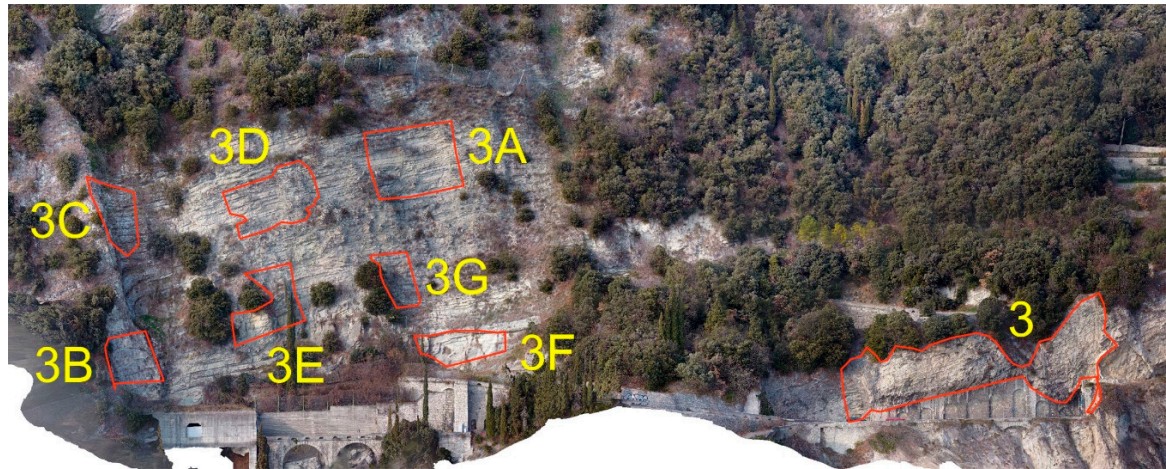

**Figure 5.** Rock face and survey windows (red polygons) analyzed in detail with non-contact surveys.

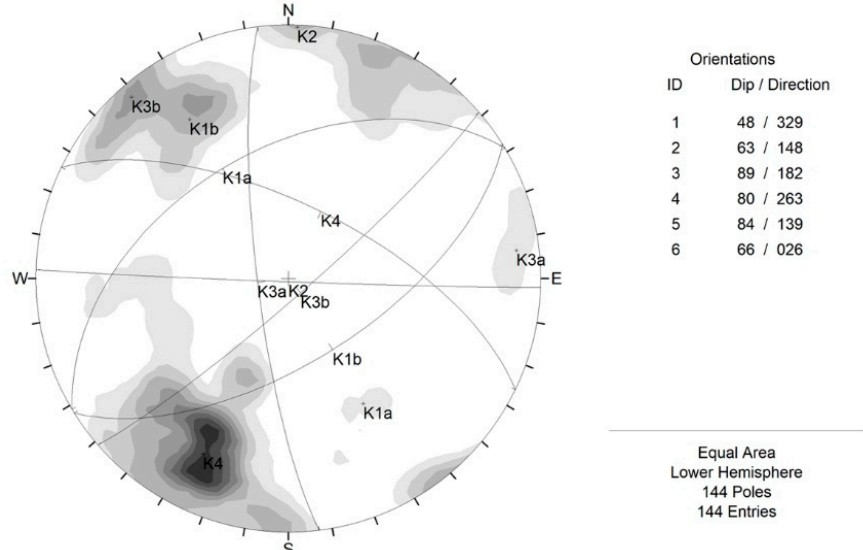

**Figure 6.** Example of a plot of a single window and the relative sets.

**Table 3.** Number of discontinuities surveyed in each window.

| Window | Number of Discontinuities |
|---|---|
| 3 | 246 |
| 3A | 282 |
| 3B | 192 |
| 3C | 176 |
| 3D | 210 |
| 3E | 146 |
| 3F | 184 |
| 3G | 144 |
| TOTAL | 1580 |

Table 4 reports the average orientation and spacing range of the main discontinuities sets obtained from non-contact surveys.

**Table 4.** Average orientation and spacing of the main discontinuity sets obtained from non-contact surveys.

| Set | Dip | Dip Direction | Spacing Range (m) |
|---|---|---|---|
| K1 | 50 | 310 | 0.07–2.87 |
| K2 | 70 | 170 | - |
| K3a | 70 | 080 | 0.08–1.60 |
| K3b | 80 | 260 | |
| K4 | 60 | 030 | 0.30–19.21 |
| K5 | 40 | 240 | 2.72–9.43 |

All sets identified by the non-contact surveys are coherent with the ones from the traditional scanline surveys. SE-dipping bedding surfaces (reported as K1b in Figure 6) due to macrofolds have been identified in sectors of the outcrops inaccessible by traditional field surveys.

The number of discontinuities of each set is extremely varied. Data belonging to K2 are very few, probably because of its orientation subparallel to the slope and/or to folded bedding. K5, despite its geometrical relationship with the face, results as the second less frequent set observed.

Spacing values among discontinuities belonging to the same set were then measured directly on the DSM, using Cloud Compare. Spacing values obtained from traditional surveys were finally added to the ones collected on the DSM and processed altogether to find the best-fitting frequency distribution for data of the entire slope.

Table 5 reports, for each set, the $p$-value of the distributions that were accepted by the K–S test applied to the spacing sample. In the case of more than one accepted distribution, the best performance corresponds to the highest $p$-value. Log-normal distribution was found to provide the best fitting of four out of five spacing samples. The number of data influences the test strongly: it is evident from the results that small samples lead to uncertainty in the identification of the best fitting distributions because the test is not able to reject the null hypothesis.

**Table 5.** The goodness of fit evaluated through the Kolmogorov–Smirnov (K–S) test (significance level $\alpha = 0.01$). Only accepted distributions are reported.

| Set | Number of Data | Distribution | $p$-Value | Best Performance |
|---|---|---|---|---|
| K1 | 266 | Gamma | 0.061 | Log-normal |
| | | Log-normal | 0.148 | |
| K2 | 12 | Gamma | 0.480 | Log-normal |
| | | exponential | 0.349 | |
| | | Log-normal | 0.718 | |
| | | Weibull | 0.488 | |
| K3 | 241 | Log-normal | 0.263 | Log-normal |
| K4 | 121 | Weibull | 0.024 | Log-normal |
| | | Log-normal | 0.078 | |
| K5 | 26 | Gamma | 0.166 | Gamma |
| | | exponential | 0.010 | |
| | | Log-normal | 0.099 | |
| | | Weibull | 0.128 | |

Table 6 reports sample arithmetic mean (SM) and distribution mean (DM) values of the total spacing sample of each set. Figure 7 shows the spacing CDF based on the best-fitting distribution.

**Table 6.** Sample arithmetic mean (SM) and distribution mean (DM) of the total spacing sample of each set.

| Set | SM [m] | DM [m] |
|-----|--------|--------|
| K1 | 0.283 | 0.293 |
| K2 | 0.463 | 0.472 |
| K3 | 0.429 | 0.434 |
| K4 | 2.020 | 2.183 |
| K5 | 3.249 | 3.611 |

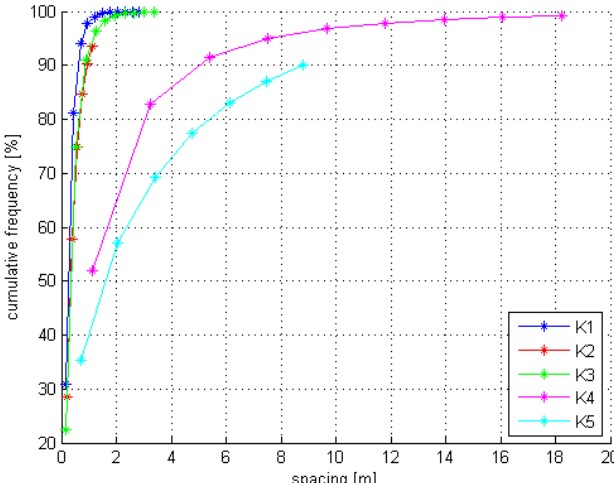

**Figure 7.** Spacing cumulative distribution function (CDF) obtained for each of the main discontinuity sets.

Among the five sets, the four more realistic combinations of three sets generating a block were considered, namely the ones that consider the strong conditioning of the bedding, the persistence of the sets K2, K4, and K5, the frequency of the K3 and the mutual geometrical relationship with the rock face. They are K1-K2-K3, K1-K2-K4, K1-K2-K5, K1-K3-K5. For the sake of simplicity, they will be referred to as terns T123, T124, T125, and T135, respectively.

Figure 8 shows IBSD calculated according to Equation (2), using coefficients $C_{i,p}$ of log-normal distribution (Table 1), and assuming fully persistent joints ($F_{imp}$ = 1). In our study, spacing values from traditional scanlines have been corrected based on scanline orientation. At the same time, measurements performed directly on the DSM do not require corrections: therefore, DM values (Table 6) match principal mean spacing ones and are input in Equation (2).

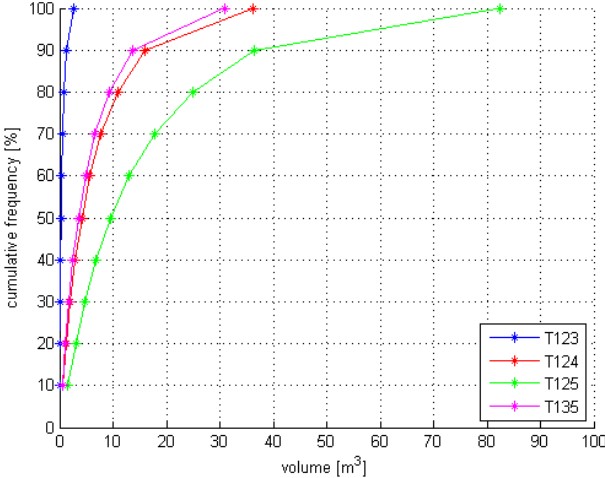

**Figure 8.** In situ block size distribution (IBSD) calculated according to Equation (2).

Figures 9 and 10 show the SCDF and MCDF, respectively, of the considered terns that create a block. Since spacing CDF of K2 and K5 are truncated at 90% (Figure 7), and at least one among them is involved in the considered terns, SCDFs are all truncated at 90% cumulative frequency. Instead, Monte Carlo simulations allow to infer the entire spacing CDFs. Therefore, MCDFs are complete.

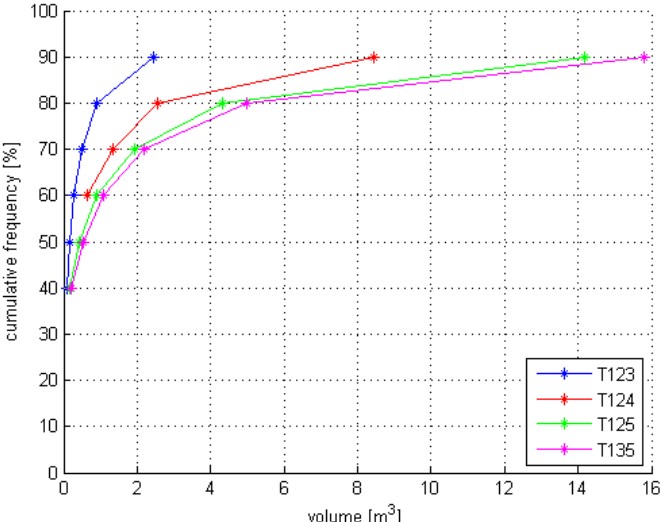

**Figure 9.** Sample CDF (SCDF) of the considered terns.

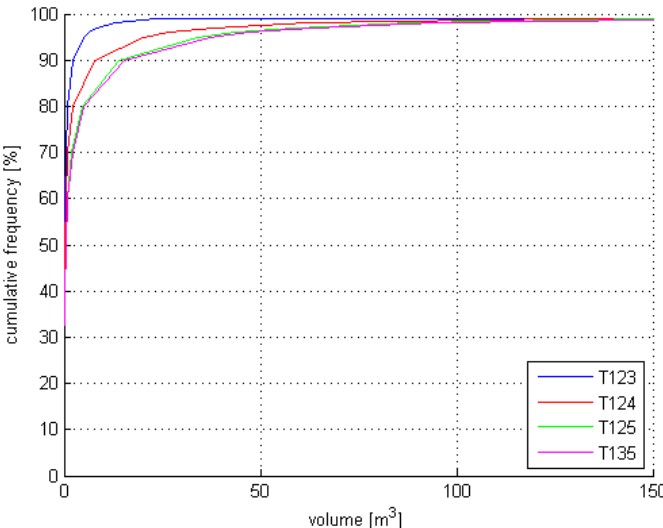

**Figure 10.** Monte Carlo CDF (MCDF) of the considered terns.

In Table 7, volumes calculated with Equation (1) considering the two different mean spacing types in Table 6 are compared: SM values produce $V_{sm}$, while DM values produce $V_{dm}$. We also show their ratios on volume correspondent to 50% cumulative frequency in MCDF ($V_{MCDF(50\%)}$), to demonstrate that Equation (1), which is generally used considering mean spacing values, does not produce a mean volume, but a value that in this case is from 2.2 to 3.6 times bigger.

Figures 11–14 show the comparison of block volume distributions (IBSD, SCDF, MCDF) obtained for the four considered terns. By definition, MCDF fits SCDF: they are both based on Equation (1) and the best-fitting distributions of spacing samples. The main utility of MCDF is to allow one to associate cumulative frequencies to volumes bigger than those covered by SDCF. Instead, IBSD nature is entirely different. In Equation (2), a unique value of spacing (DM) is used for each set of the considered tern. Therefore, a sort of "base volume" is calculated, and the coefficients of Table 1 are multiplied to the

same "base volume" to build the IBSD. It can be observed that IBSD intersects SCDF and/or MCDF in all the four cases. The agreement among the curves is at its maximum for T123. For the other terns, IBSD tends to produce bigger volumes with the same cumulative frequency, until the intersection is reached.

**Table 7.** Comparison of ratios between volumes calculated considering different mean spacing types.

| Tern | $V_{sm}$ [m³] | Ratio $V_{sm}/V_{MCDF(50\%)}$ | $V_{dm}$ [m³] | Ratio $V_{dm}/V_{MCDF(50\%)}$ |
|------|---------------|-------------------------------|---------------|-------------------------------|
| T123 | 0.28 | 2.17 | 0.29 | 2.32 |
| T124 | 0.78 | 3.17 | 0.88 | 3.57 |
| T125 | 1.21 | 2.91 | 1.41 | 3.40 |
| T135 | 1.35 | 2.92 | 1.55 | 3.37 |

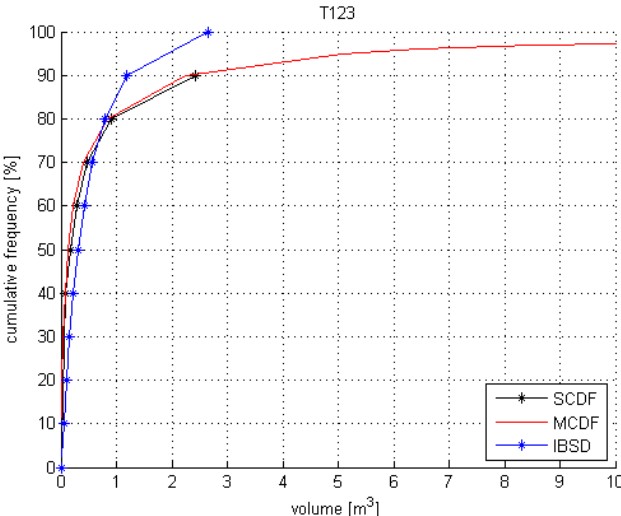

**Figure 11.** Comparison of block volume distributions obtained for term T123.

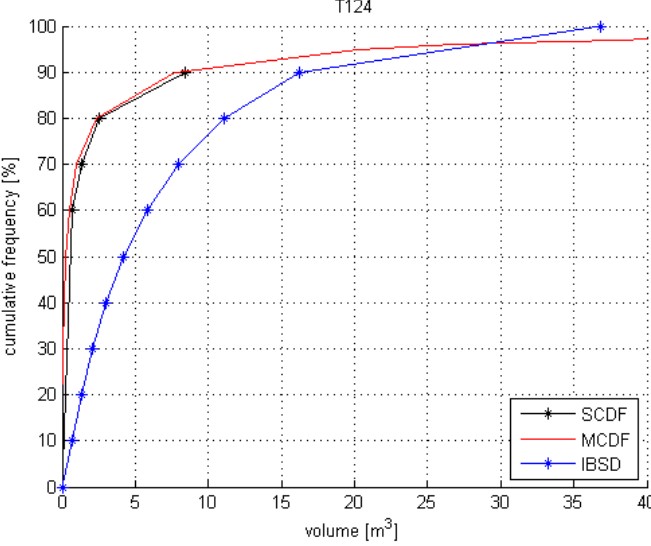

**Figure 12.** Comparison of block volume distributions obtained for term T124.

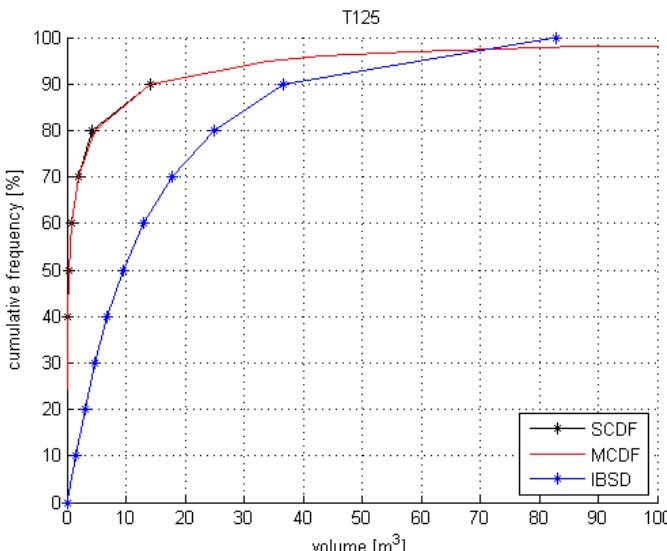

**Figure 13.** Comparison of block volume distributions obtained for term T125.

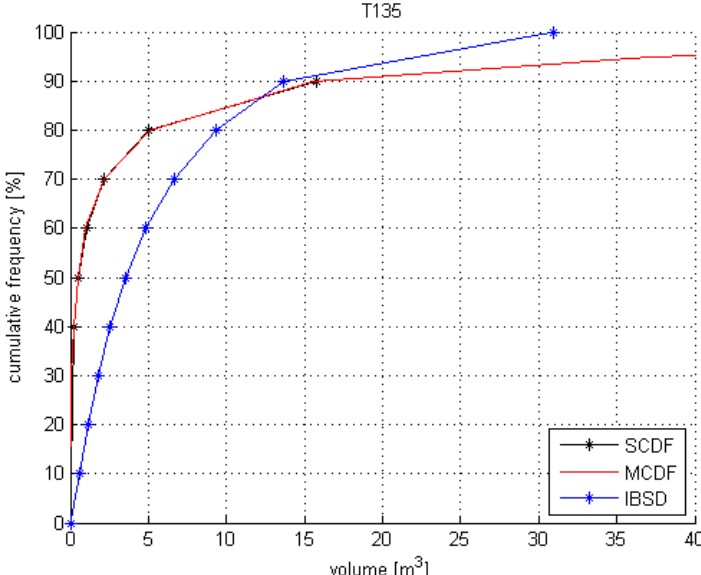

**Figure 14.** Comparison of block volume distributions obtained for term T135.

Figure 15A represents one of the images used for measuring blocks at the foot of the slope with ImageJ methodology. Figure 15B shows the frequency of detached blocks (SV) for the two assumed shapes (Equations (4) and (5)).

Table 8 reports, for the considered terms, the ratios between the volume correspondent to 100% cumulative frequency in IBSD and the maximum surveyed volume ($SV_{max}$), and between the 95% cumulative frequency in MCDF ($V_{MCDF(95\%)}$) and $SV_{max}$, considering the two assumed shapes (parallelepiped and ellipsoid). The ratio is an indicator of the representativity of surveyed volumes with respect to the distribution obtained from spacing values. It is evident that in this case, blocks at the foot of the slope are some order of magnitude smaller than a plausible theoretical maximum value.

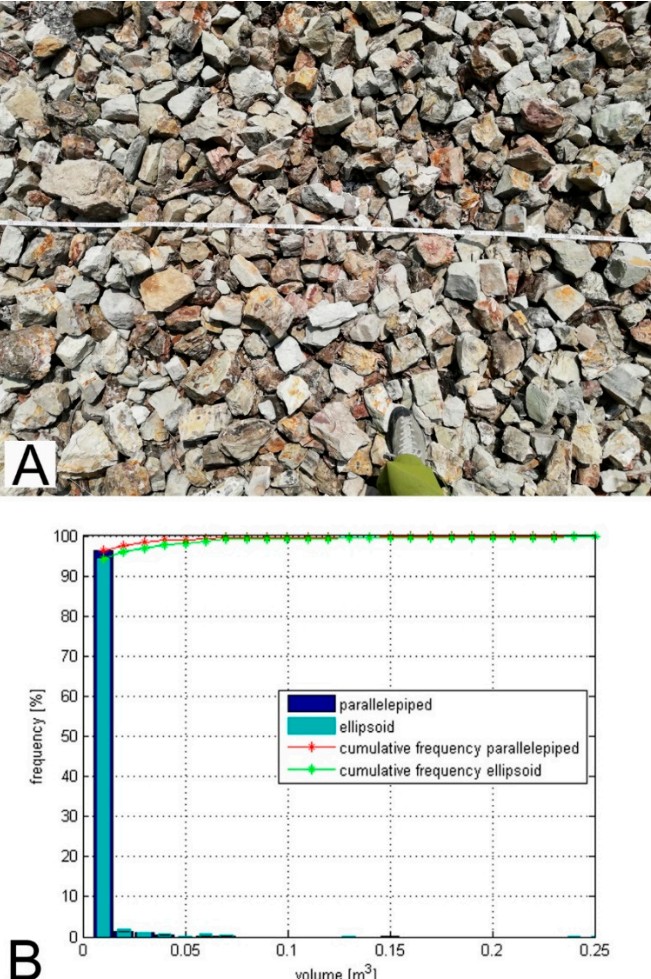

**Figure 15.** (**A**) One of the images processed with ImageJ for surveying blocks at the foot of the slope. (**B**) Frequency of detached blocks (SV), considering the two assumed shapes.

**Table 8.** Ratios calculated for the considered terms.

| Tern | Ratio $V_{IBSD(100\%)}/SV_{max}$ [-] | | Ratio $V_{MCDF(95\%)}/SV_{max}$ [-] | |
|------|----------------|-----------|----------------|-----------|
|      | Parallelepiped | Ellipsoid | Parallelepiped | Ellipsoid |
| T123 | 17.7 | 11.2 | 33.5 | 21.1 |
| T124 | 243.3 | 153.4 | 134.4 | 84.7 |
| T125 | 549.5 | 346.4 | 223.4 | 140.8 |
| T135 | 204.5 | 128.9 | 242.9 | 153.1 |

## 5. Discussion

A few main considerations derive from the application of traditional and non-contact methods to the same rockface.

The ranges of spacing and persistence values estimated for the same set with traditional and non-contact survey techniques are different (Tables 2 and 4). This difference depends on the geological features of the analyzed rock succession and the limit of detection of the non-contact survey.

The bedding K1 identifies the most pervasive discontinuity, and its features depend on the depositional history of the sedimentary succession, leading to changes in composition, texture, and thickness from strata to strata. The K1 spacing defined directly on the outcrops by the traditional approach is generally lower than the values obtained by the application of the non-contact method. The field study of the lithological multilayer estimated average spacing values in the order of few

centimeters, with marls usually from thin to very thin bedded and limestone beds only a few centimeters thick (Figure 16A).

In comparison, the non-contact surveys have provided an average value of about 20 cm. Thus, rocky volumes comprised between two K1 surfaces drawn by non-contact surveys can be actually more bedded. Besides, K1 spacing could be decreased by possible structural discontinuities subparallel to the bedding due to the regional tectonics.

Moreover, field observations show that pervasive K3 discontinuities, crossing the bedding surfaces, are significantly branched by the lithological multilayer consisting of limestone and marls with contrasting competence (Figures 2B and 16B).

The spacing of the high angle discontinuity sets (K4 and K5), being of metric and plurimetric order, was better measured using CloudCompare directly on the DSM of the slope. In fact, in the traditional surveys, due to the spacing values that characterize them, they are poorly represented or not at all. They appear either as random planes or with too few measurements to be able to derive reliable mean values statistically.

K4 and K5 discontinuities are crucial in the rock face stability: vast portions of the rock mass have been observed to be mainly isolated by the intersection of these discontinuities with the other surfaces (Figure 16C). Thus, it is likely to assume that such large volumes tend to subdivide into smaller blocks already at the time of detachment (due to the above reasons), and then, they may further reduce in size as a result of impacts during the fall.

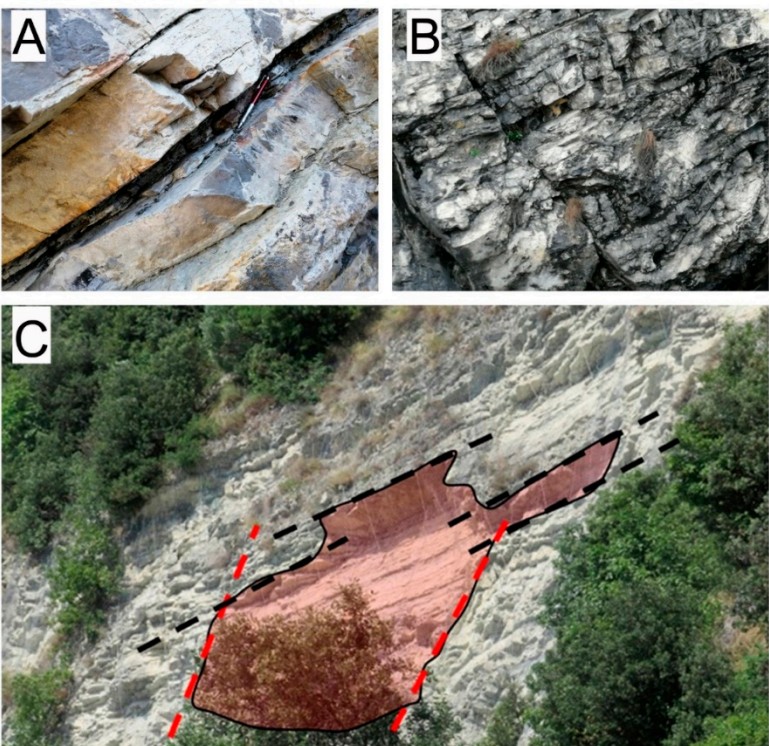

**Figure 16.** (**A**) Marly layers and bedding planes in limestones often not detectable by non-contact methods. (**B**) High-angle discontinuities fracturing bedded limestones and contributing to the small size of blocks observed onsite. (**C**) A big volume of rock mass isolated by the intersection of the most spaced discontinuity sets with the bedding.

This fact would justify the small size (values below 0.25 m$^3$) of the blocks regularly measured at the foot of the slope, which is smaller than the ones obtained by data from traditional and non-contact surveys. Nevertheless, the volumes of detached blocks are huge in comparison to the size of single blocks due to the influence and mutual interaction of persistent and high-spaced sets (K2-K4-K5).

The volumes of the blocks on the ground could correspond then to small unitary rocky volumes, initially forming large masses isolated by the discontinuities with high spacings. Probably, they were further fragmented by the impacts during the fall, especially given the marly component that reduces the strength of the intact rock, and by the occurrence of bedding surfaces, also at a minor scale.

In light of the above observations, a statistically robust block volume distribution assumes a fundamental role in covering the entire possible volume range and associating to each value a probability of not being exceeded. Each spacing sample associated with a discontinuity set should contain a statistically sufficient number of measurements and be representative of its variability in the considered rock mass. The larger the maximum spacing, the greater the considered area must be to allow enough measurements to be made. The use of non-contact survey methods is crucial in this sense. In any way, traditional surveys cannot be neglected: their sensibility in the identification of the peculiar features of discontinuities and the associated structural elements is irreplaceable.

A robust fitting process and goodness-of-fit evaluation are also fundamental for selecting the most suitable distributions for describing spacing samples and build the proper CDFs.

The comparison of the tested block volume cumulative distributions obtained for the four considered terns shows a good agreement. MCDF is an extension of SCDF: the more complete the SCDF, the more accurate the MCDF. On this basis, MCDF allows to associate cumulative frequencies to volumes bigger than those covered by SDCF. In general, with the same cumulative frequency, we obtained IBSD volumes bigger than MCDF ones until the intersection with MCDF is reached.

Block volumes surveyed at the foot of the slope, in this case, yield data regarding only the smallest possible blocks. Visual inspection of rockfall scars (Figure 16C) confirms this fact. Since just the first part of the IBSD and MCDF overlap SV (Table 8), it is impossible to perform a real validation of the obtained distributions. Moreover, it is not possible to indicate which is the contribution of each considered tern in the total block volume distribution. It is the opinion of the authors that the ongoing research regarding the creation of a connection between the IBSD and the RBSD, considering the fragmentation process [57,58], is of fundamental importance and requires contributions from the scientific community.

## 6. Conclusions

This paper deals with the definition of ISBD for a highly fractured rock mass outcropping along the western Lake Garda (Italy). Different methodologies for the definition of design block volumes were discussed, and their results were compared, highlighting strengths and limitations.

Data from in situ traditional and non-contact surveys were analyzed for identifying the main discontinuity sets and their characteristics in terms of orientation and spacing. Statistical analyses on spacing values were performed for evaluating the frequency distribution of each discontinuity set. Log-normal and Gamma distributions demonstrated capability of simulating frequency distribution for discontinuity set spacing. However, the procedure described in this paper is subordinate to data availability (in terms of quality and quantity of data): the combination of non-contact and traditional surveys allows for this type of analysis, and consequently, the statistical results can be considered robust and reliable.

One of the main findings of this paper is the huge discrepancy between RBSD and ISBD results: the volume of fallen blocks distributed along the base of the rock face is far smaller than that estimated from spacing analysis on the rock mass. Consequently, if an RBSD value was used for barrier design, it would lead to rough mistakes and an underestimation of impacting energy.

In highly fractured rock masses, such as those here described, the fallen rock blocks have a marginal role in design block determination, since their volume could be affected by other processes after the detachment (e.g., fragmentation). Conversely, stratigraphy and structure play a fundamental role, since they mirror the degree of fracturing of the rock face. However, an accurate field observation, i.e., stratigraphy and structures of the rock face, is imperative for supporting any approaches to study rockfall phenomena.

The procedure here described should be standard practice in the study of rockfall events, and it should be uniform in European standards such as Eurocodes. Future developments should involve the scientific community for setting the percentiles of the probability distribution to be considered for the design block definition.

**Author Contributions:** Conceptualization, G.U. and S.M.R.B.; Data curation, G.U., S.M.R.B., and P.M.; Formal analysis, G.U. and S.M.R.B.; Funding acquisition, A.M.F.; Investigation, P.M.; Methodology, G.U., S.M.R.B., and P.M.; Project administration, S.M.R.B. and A.M.F.; Resources, G.U., S.M.R.B. and P.M.; Software, G.U.; Supervision, A.M.F.; Validation, G.U., S.M.R.B., and P.M.; Visualization, G.U., S.M.R.B., and P.M.; Writing—original draft, G.U., S.M.R.B., P.M., and F.V.; Writing—review and editing, G.U. and F.V. All authors have read and agreed to the published version of the manuscript.

**Funding:** This research received no external funding.

**Acknowledgments:** The authors thank IMAGEO S.r.l, who performed the laser scanner and photogrammetric surveys and provided images and DSM used in this study. The authors also thank Manuel Bertholin, who contributed to this study as part of his Master's Degree thesis.

**Conflicts of Interest:** The authors declare no conflict of interest.

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
