# Peer review of "In Situ Block Size Distribution Aimed at the Choice of the Design Block for Rockfall Barriers Design: A Case Study along Gardesana Road"

_geosciences, doi:10.3390/geosciences10060223_

Round 1
Reviewer 1 Report
The authors present a complete review of methods to determine the ISBD with the aim to propose simple approaches that could be implemented by practitioners interested in modelling the propagation of rockfall masses and in the design of protective measures. The authors have developed a procedure for obtaining the IBSD from the discontinuity data collected with the use of both traditional (scanlines) and non-contact techniques (photogrammetry). The paper is well organized and the conclusions are well supported by the data. Therefore, I consider that the manuscript deserves to be published without additional changes.
Author Response
We thank the reviewer very much for his/her positive comment.
Best regards.
Reviewer 2 Report
Congratulations, you performed an interesting study that can contribute to improving European Standards. For instance, a relevant finding is the significant mistakes that could be occurred if values from RBSD are used as a design block for barrier design to avoid physical damages due to rockfall events.
Author Response

(The authors gave the same response as above.)

Reviewer 3 Report
Please look at the corrections and suggestions in the attached file.

Author Response
We thank the reviewer very much for his/her comments and suggestions for language improvement.
We corrected the text according to the suggestions. We rewrote the highlighted sentences, as follows.
Line 127 “The exposed sedimentary succession is part of the Lombardian Basin, tectonically overlain by Norian platform carbonates to the west”.
Lines 139-141 “In this area, open to tight asymmetrical folds (Fig. 2A) have mainly NE-SW trending axes and axial planes variably dipping towards NW.”
Line 155-157 “In the area, mesoscopic compressional structures (top-to-SE) subparallel to the bedding have been observed.”
Line 191 we added the definition of principal mean spacing according to Lu & Latham. “… the principal mean spacing values, namely the average true spacing values of the three sets; …”
Line 312-313 we deleted the sentence, because it was useless.
In addition, we replaced Figure 1 with a new figure in which Lake Garda is used instead of Garda Lake. We modified in the manuscript accordingly.
